# Modeling *Aceria tosichella* biotype distribution over geographic space and time

**Luaay Khalaf[1,2], Alicia Timm[3], Wen-Po Chuang[4], Laramy Enders[5], T. J. Hefley[6], C. Michael Smith[1]***

**1** Department of Entomology, Kansas State University, Manhattan, Kansas, United States of America,
**2** Department of Plant Protection, College of Agriculture, University of Baghdad, Baghdad, Iraq,
**3** Department of Bioagricultural Sciences and Pest Management, Colorado State University, Fort Collins, Colorado, United States of America, **4** Department of Agronomy, National Taiwan University, Taipei, Taiwan,
**5** Department of Entomology, Purdue University, West Lafayette, Indiana, United States of America,
**6** Department of Statistics, Kansas State University, Manhattan, Kansas, United States of America

\* cmsmith@ksu.edu

**Data Availability Statement:** The nucleotide sequences of ITS1 region and COI used in phylogenetic analyses have been deposited in GenBank (accession numbers MT336812-

## Abstract

The wheat curl mite, *Aceria tosichella* Keifer, one of the most destructive arthropod pests of bread wheat worldwide, inflicts significant annual reductions in grain yields. Moreover, *A. tosichella* is the only vector for several economically important wheat viruses in the Americas, Australia and Europe. To date, mite-resistant wheat genotypes have proven to be one of the most effective methods of controlling the *A. tosichella*—virus complex. Thus, it is important to elucidate *A. tosichella* population genetic structure, in order to better predict improved mite and virus management. Two genetically distinct *A. tosichella* lineages occur as pests of wheat in Australia, Europe, North America, South America and the Middle East. These lineages are known as type 1 and type 2 in Australia and North America and in Europe and South America as MT-8 and MT-1, respectively. Type 1 and type 2 mites in Australia and North America are delineated by internal transcribed spacer 1 region (ITS1) and cytochrome oxidase I region (COI) sequence differences. In North America, two *A. tosichella* genotypes known as biotypes are recognized by their response to the *Cmc3* mite resistance gene in wheat. *Aceria tosichella* biotype 1 is susceptible to *Cmc3* and biotype 2 is virulent to *Cmc3*. In this study, ITS1 and COI sequence differences in 25 different populations of *A. tosichella* of known biotype 1 or biotype 2 composition were characterized for ITS1 and COI sequence differences and used to model spatio-temporal dynamics based on biotype prevalence. Results showed that the proportion of biotype 1 and 2 varies both spatially and temporally. Greater ranges of cropland and grassland within 5000m of the sample site, as well as higher mean monthly precipitation during the month prior to sampling appear to reduce the probability of occurrence of biotype 1 and increase the probability of occurrence of biotype 2. The results suggest that spatio-temporal modeling can effectively improve *A. tosichella* management. Continual integration of additional current and future precipitation and ground cover data into the existing model will further improve the accuracy of predicting the occurrence of *A. tosichella* in annual wheat crops, allowing producers to make informed decisions about the selection of varieties with different *A. tosichella* resistance genes.

MT337241 for ITS1 region, and MT370025-
MT370073 for COI).

**Funding:** This work was supported by USDA/NIFA
North Central IPM Program award 2013-34103-
21208 to CMS, the Kansas Wheat Alliance, funding
by the University of Baghdad to LK, and Kansas
State University Research and Extension. The
funders had no role in study design, data collection
and analysis, decision to publish, or preparation of
the manuscript.

**Competing interests:** The authors have declared
that no competing interests exist.

# Introduction

The wheat curl mite, *Aceria tosichella* Keifer, is a global pest of bread wheat *Triticum aestivum*
L. The mite reduces grain yield by causing both direct feeding damage and as a vector of several viral wheat pathogens [1–7]. Yield losses caused by *A. tosichella* feeding may be up to 30%
[4, 8] due to leaf rolling and trapping [9]. *Aceria tosichella* transmits three damaging viruses to
the wheat plant—*Wheat Streak Mosaic Virus* (WSMV, family Potyviridae, genus *Tritimovirus*),
*High Plains wheat mosaic virus* (HPWMoV, genus *Emaravirus*, formerly *High plains virus*;
www.ictvonline.org/proposals-15/2015.018aP.A.v3.Emaravirus_sp.pdf), and *Triticum Mosaic
Virus* (TriMV, family Potyviridae, genus *Poacevirus*). *Aceria tosichella* nymphs obtain WSMV
after feeding for as little as 30 min on infected plants and can spread the virus for at least 7d
postfeeding. [9–13]. Although WSMV infections occur at a greater incidence than HPWMoV
or TriMV [14], co-infections are common [15–17]. WSMV causes wheat yield losses ranging
from 2.5 to 7% on at least five different continents [18] depending on climate, virus acquisition
time and wheat cultivar [13, 19–24]. *Aceria tosichella* detection is complicated by the mite's
small size (150–225 μm length), ability to attain maximum concealment through cryptic
behavior and wide host range [10, 18, 25–31]. To date, no effective acaricides exist to manage
*A. tosichella* and the viruses it transmits [32, 33]. The cultural practice of controlling over-summering hosts such as volunteer wheat and weed grass hosts can provide effective management
of the *A. tosichella*-virus complex if producers use this management approach [34].

  *Aceria tosichella* is a complex global mixture of at least 29 different genetic lineages [28, 30,
31]. However, two genetically distinct lineages occur as pests of wheat in Australia, Europe,
North America, South America and the Middle East. These lineages are known as type 1 and
type 2 in Australia and North America and in Europe and South America type 1 and type 2 are
known as MT-8 and MT-1, respectively [28]. In North America, internal transcribed spacer 1
region (ITS1) sequence differences were used to delineate two lineages in mites collected in
Kansas, Montana, Alberta Canada, and Nebraska [35]. More recently, type 1 and type 2 lineages have been delineated using ITS1- and cytochrome oxidase I region (COI) sequence differences [29, 36]. Evidence also exists to show that the *A. tosichella* type 1 and type 2 lineages
differ in their ability to transmit WSMV, HPWMoV and TriMV in Australia and North America [37, 38, 39].

  The development and use of *A. tosichella*-resistant wheat cultivars to reduce *A. tosichella*
populations and WSMV infection has progressed in North America since 1995 [40–45]. During this process, North America *A. tosichella* populations also began to be referred to as biotypes because of the ability of one biotype to overcome the effect of (exhibit virulence to) *A.
tosichella* resistance gene(s) in wheat. Currently, biotype 1 is referred to as being avirulent to
the effects of the rye:wheat translocation resistance gene (*Cmc3* [curl mite colonization]) in the
wheat variety TAM 107. Biotype 2 is recognized as being virulent to *Cmc3* [46].

  Virulence in *A. tosichella* to *Cmc3* has remained stable for the past 20 years [34] but recent
field assessments [47] determined that 24% of *A. tosichella* collected from multiple locations in
North America are virulent to *Cmc3*. Therefore, there is a real need for new information about
the current geographic distribution of *A. tosichella* biotypes or genetic lineages throughout the
U. S. Great Plains and the potential changes occurring in each. In order to obtain new knowledge for more effective *A. tosichella* management programs, a regional study was conducted to
assess the current genetic variation of *A. tosichella*. Our hypothesis was that changes in *A. tosichella* genetic composition are dynamic and have spatial and temporal structure that may be
correlated with climate and landscape features. To test this hypothesis, experiments were conducted to assess the geographic distribution of *A. tosichella* in six U. S. Great Plains wheat-producing states in 2014 and 2015 based on internal transcribed spacer 1 (ITS1) region and

cytochrome oxidase I (COI) polymorphisms and plant phenotypic reactions. An additional experiment was conducted to compare in-depth sequence analyses of *A. tosichella* populations at four locations in Kansas, Missouri and Nebraska in 2016 to determine variation over local scales. Finally, temporal variation in *A. tosichella* lineages over a 2-year period was used to develop a generalized additive spatio-temporal model to predict the prevalence of biotypes 1 and 2 in the Great Plains.

## Results

### Phylogenetic analyses

A total of 430 *A. tosichella* were collected in 2014, 2015, and 2016 from 12 locations in 2014, 13 locations in 2015, and 12 locations in 2016 (S1 and S2 Tables). These samples yielded regions of 618 bases for ITS1 analysis in all samples and 506 bases for COI analysis in 49 samples. All mites analyzed were *A. tosichella* [29, 36]. Bayesian phylogenetic analyses revealed clearly distinct differences between biotypes, based on 8 ITS1 haplotypes and 3 COI haplotypes. Genbank sequences that were used for comparison confirmed biotype 1 and 2 designations, as well as MT-8 and MT-1 designations [48]. Genetic distance values between haplotypes based on ITS1 polymorphism ranged from 0.003 to 0.028 (S3 Table), whereas these values ranged from 0.002 to 0.177 based on COI polymorphism (S4 Table).

### *Aceria tosichella* biotype distribution

Biotypes 1 and 2 were present in 2014 and 2015 in all sampling sites with the exception of six fields in Missouri and two fields in Texas, where only biotype 2 was present (Fig 1A). Biotype 1 occurred as five haplotypes and biotype 2 was present as three haplotypes. One biotype 1 haplotype with a 1 base pair (bp) difference to the primary haplotype was observed in about one third of the populations and was present primarily in Kansas and Missouri. Although biotype 2 was present in six states (Fig 1A), most of the variation in this biotype was observed in Kansas and Texas. Interestingly, a new biotype 1 haplotype differing by 1bp to the dominant haplotype appeared in all 3 years of sampling (Fig 1A and 1B).

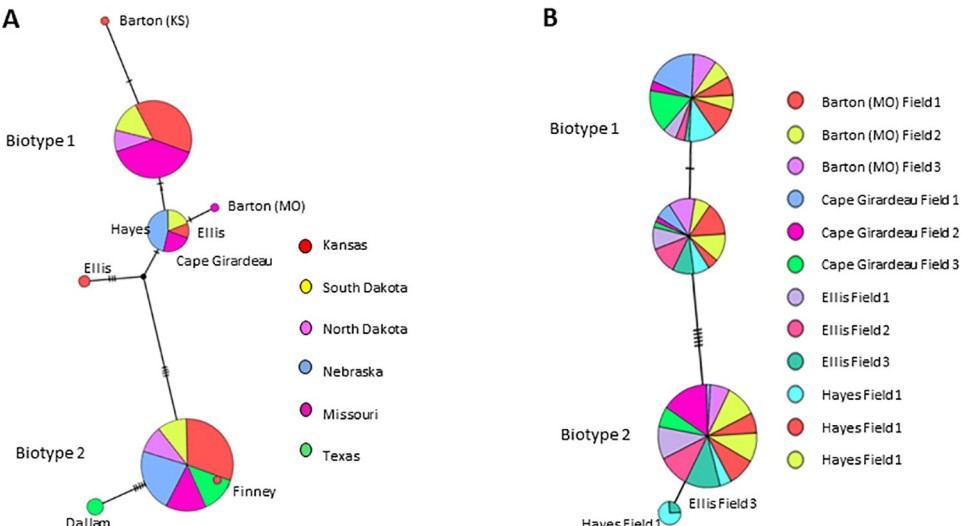

**Fig 1. Phylogeny of *A. tosichella* haplotypes sampled in 2014 and 2015.** (A) and 2016 (B) within biotypes 1 and 2. Smaller circles indicate fewer individuals in a haplotype. Hash marks on lines connecting haplotypes symbolize base-pair differences.

**Table 1. Proportion of *A. tosichella* biotype 1 and 2 in samples collected in Kansas, Missouri, Nebraska, North Dakota, South Dakota and Texas in 2014 and 2015 (n = 10 per county).**

| State | County | Proportion of biotype 1:2 |
|---|---|---|
| Kansas (n = 80) | Saline | 6:4 |
| | Geary | 9:1 |
| | Greeley | 4:6 |
| | Dickinson | 3:7 |
| | Barton | 5:5 |
| | Finney | 4:6 |
| | Ellis | 7:3 |
| | Ellsworth | 2:8 |
| | Average | 5:5 |
| Nebraska (n = 40) | Cheyenne | 3:7 |
| | Hayes | 4:6 |
| | Furnas | 0:10 |
| | Saunders | 0:10 |
| | Average | 2:8 |
| Missouri (n = 60) | Barton | 8:2 |
| | Cape Girardeau | 9:1 |
| | Pike | 7:3 |
| | Pettis | 8:2 |
| | Stoddard | 6:4 |
| | Cooper | 4:6 |
| | Average | 7:3 |
| North Dakota (n = 20) | Ward | 5:5 |
| | Bottineau | 3:7 |
| | Average | 4:6 |
| South Dakota (n = 30) | Hughes | 7:3 |
| | Tripp | 8:2 |
| | Lake | 2:8 |
| | Average | 5:5 |
| Texas (n = 20) | Randall | 0:10 |
| | Dallam | 0:10 |
| | Average | 0:10 |

Each biotype was present at all sample locations during the study, with the exception of two locations in Nebraska and two locations in Texas in 2014 and 2015 (Table 1). Individual grain head samples obtained in 2016 from Missouri, Nebraska and Kansas also contained both biotypes Table 2). However, biotype 1 occurred in greater frequency in Missouri locations, where the average ratio of mites sampled at six locations was 70% biotype 1 and 30% biotype 2 (Table 1). A higher probability of occurrence of biotype 2 occurred at four Nebraska locations (20% biotype 1, 80% biotype 2; and at two Texas locations that each yielded 100% biotype 2 (Table 1). Biotype ratios were evenly distributed in Kansas and South Dakota (50% for each biotype); and in North Dakota (40% biotype 1, 60% biotype 2) (Table 1).

Mites collected from individual wheat heads in fields in Kansas, Missouri and Nebraska during 2016 were mixtures of both biotypes, indicating that both occur on a single grain head simultaneously (Fig 2). Of 36 heads examined, 28 contained both biotypes. However, eight heads contained a single biotype. Seven of these eight heads contained only biotype 1 and occurred in Barton County Missouri (field 3, head 2); Cape Girardeau County Missouri (field

**Table 2. Ratios of *A. tosichella* biotype 1 and 2 at one location in Kansas, two locations in Missouri, and one location in Nebraska in 2016.** A total of three fields at each location were sampled, five individuals were collected at each of three sites in each field for a total of 45 individuals per field.

| State | County | Field #. Sample site # | Ratio of biotype 1:2 |
|---|---|---|---|
| Kansas | Ellis | 1.1 | 6:4 |
| | | 1.2 | 2:8 |
| | | 1.3 | 6:4 |
| | | 2.1 | 6:4 |
| | | 2.2 | 6:4 |
| | | 2.3 | 2:8 |
| | | 3.1 | 2:8 |
| | | 3.2 | 2:8 |
| | | 3.3 | 6:4 |
| | | Average | 4:6 |
| Missouri | Barton | 1.1 | 6:4 |
| | | 1.2 | 8:2 |
| | | 1.3 | 6:4 |
| | | 2.1 | 4:6 |
| | | 2.2 | 6:4 |
| | | 2.3 | 4:6 |
| | | 3.1 | 4:6 |
| | | 3.2 | 10:0 |
| | | 3.3 | 6:4 |
| | | Average | 6:4 |
| Missouri | Cape Girardeau | 1.1 | 8:2 |
| | | 1.2 | 10:0 |
| | | 1.3 | 10:0 |
| | | 2.1 | 4:6 |
| | | 2.2 | 6:4 |
| | | 2.3 | 10:0 |
| | | 3.1 | 10:0 |
| | | 3.2 | 0:10 |
| | | 3.3 | 10:0 |
| | | Average | 8:2 |
| Nebraska | Hayes | 1.1 | 8:2 |
| | | 1.2 | 4:6 |
| | | 1.3 | 6:4 |
| | | 2.1 | 4:6 |
| | | 2.2 | 10:0 |
| | | 2.3 | 2:8 |
| | | 3.1 | 6:4 |
| | | 3.2 | 6:4 |
| | | 3.3 | 4:6 |
| | | Average | 5.5 |

1 heads 2 and 3, field 2 head 3; field 3 heads 1 and 3) and in Hayes County Nebraska (field 2 head 2) (Table 2) (Fig 3A and 3B). In a single instance, only biotype 2 was found on head 2 collected in Cape Girardeau County Missouri field 3 (Fig 3C).

Samples of individual grain heads collected in 2016 in Nebraska, Kansas and Missouri revealed some changes in biotype ratios from those in 2015. In Hayes County Nebraska the

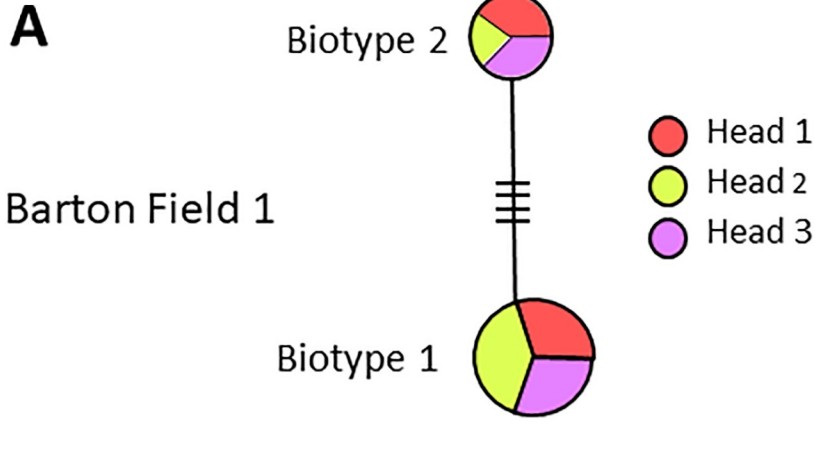

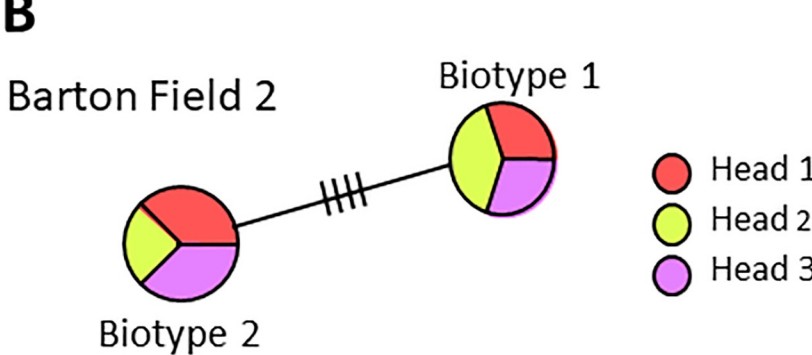

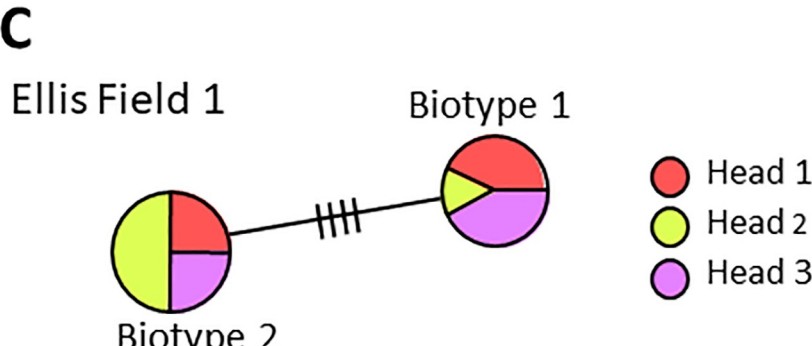

**Fig 2. Phylogeny of *A. tosichella* haplotypes sampled in 2016 within biotypes 1 and 2.** (A) Barton county Missouri field 1 and (B) field 2; and (C) Ellis County Kansas field 1. Each field contained biotype 1 and biotype 2 in all heads sampled. Smaller circles indicate fewer individuals in a haplotype. Hash marks on lines connecting haplotypes symbolize base-pair differences.

ratio changed slightly from 4:6 in 2015 to 5:5 in 2016; and in Cape Girardeau County Missouri, the ratio remained primarily biotype 1, shifting from 9:1 to 8:2 (Tables 1 and 2). Similarly, the ratio shifted slightly from 8:2 in 2015 to 6:4 in 2016 in Barton County Missouri. However, the ratio in Ellis County Kansas shifted significantly from 7:3 in 2015 to 4:6 in 2016 (Tables 1 and 2).

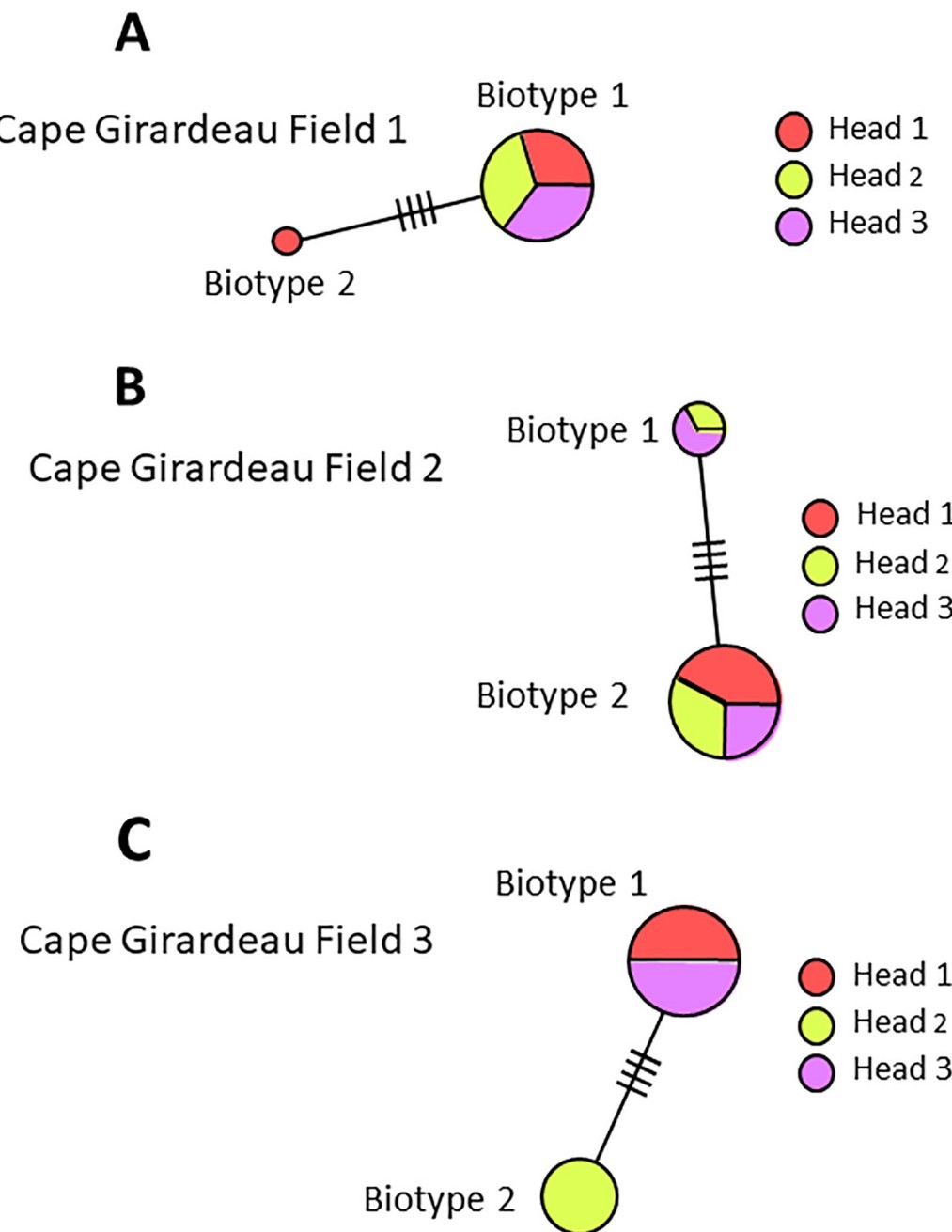

**Fig 3. Phylogeny of *A. tosichella* haplotypes sampled in 2016.** (A) Cape Girardeau county Missouri field 1 containing biotype 1 in all heads and biotype 2 in one head only; (B) Cape Girardeau county Missouri field 2 containing biotype 1 in two heads and biotype 2 in all heads; (C) Cape Girardeau county Missouri field 3 containing biotype 1 in two heads and biotype 2 in one head only. Smaller circles indicate fewer individuals in a haplotype. Hash marks on lines connecting haplotypes symbolize base-pair differences.

## Predicted *A. tosichella* biotype occurrence

The probability of occurrence of each biotype showed distinct spatio-temporal patterns, which were influenced by predictor variables (Fig 4). The percentage of grass/pasture and crop land within 5000 m of a sample location had negative coefficient estimates with 90% confidence

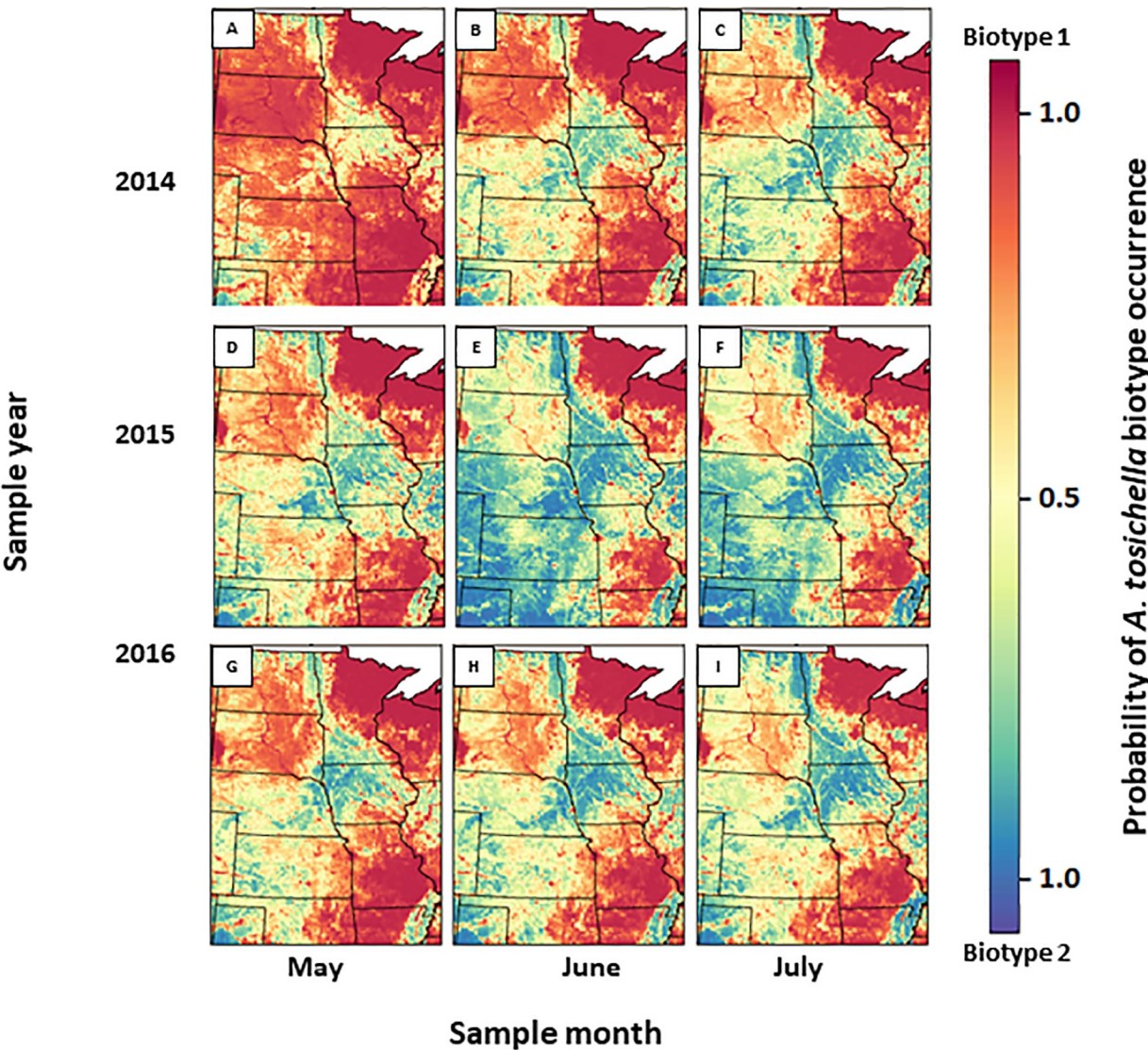

**Fig 4. Predictive spatio-temporal heatmaps of the expected probability of *A. tosichella* biotype 1 and 2 in May, June and July of 2014, 2015 and 2016 as affected by increased precipitation 30 d prior to sampling and grass/pasture:cropland landcover ratio.** Maps do not consider specific agricultural landscape and can be interpreted as the potential number of biotype 1 or 2 expected if a given map location was wheat. Red areas—higher probability of biotype 1 presence; blue areas—higher probability of biotype 2 presence. Note: These data were collected during the periods May 21 to July 10, 2014; June 25 to July 12, 2015; and June 245 11 to June 16, 2016. As a result, the predictive maps are an extrapolation if used to infer dynamics outside of the time periods in which these data were collected.

intervals (CI) that did not include zero. Negative coefficient estimates indicate that as the predictor variable increases (e.g., percentage of crop land with 5000 m) the probability of an *A. tosichella* individual being biotype 1 decreases. Similar to the land cover predictor variables, precipitation during the month of sampling and the month prior to sampling had negative coefficient estimates, but in this case the upper limit of the 90% CI was slightly greater than zero, indicating less certainty regarding the sign and magnitude of the effects. All other climate predictors (i.e., mean monthly temperature) and landscape effects (i.e., the interaction of crop land and grass/pasture) had 90% CI that indicated a substantial level of uncertainty about the sign and magnitude of the effect. We refrain from interpreting the impact of these variables on

the occurrence of the biotypes. In summary, our results demonstrate that increases in crop and grass/pasture within 5000 m of the sample location along with monthly precipitation during the month prior to and month of sampling reduced the probability of occurrence of biotype 1, and consequently increased the probability of occurrence of biotype 2 in some locations in Nebraska, North Dakota, South Dakota, and Kansas. Fig 4 shows the expected probability of each biotype at each sample location assuming a given point was wheat and that mites were present at that location.

The spatio-temporal statistical model predicted that the probability of occurrence of biotype 1 was particularly high at the majority of dates and locations in Missouri and South Dakota in 2014, 2015 and 2016 (Fig 4, S1A–S1E and S2A–S2C Figs). In contrast, the model predicted a decreased probability of occurrence of biotype 1 in 2014 and 2015 in North Dakota and a gradual increase of biotype 1 probability by the end of 2016 (Fig 4, S3A and S3B Fig). An even greater decrease in the probability of occurrence of biotype 1 was predicted at the majority of locations in Kansas and Nebraska (Fig 4B–4I). For example, predicted probability of biotype 1 occurrence was low at most locations in Kansas and Nebraska in 2015 (Fig 4D–4F), but increased to a nearly equal probability of occurrence of each biotype in 2016 (Fig 4G–4I, S4A–S4C and S5 Figs) with the exception of the eastern-most sample location in Nebraska, where biotype 2 was predicted to be predominant in 2015 and 2016 (Fig 4D–4I, S4D Fig). Finally, the model predicted biotype 2 occurrence at the two Texas locations in all three sample years (Fig 4, S5 Fig).

## Discussion

Sequencing ITS1 and COI polymorphisms in *A. tosichella* samples obtained in the current study indicate that these genes remain useful genomic regions for *A. tosichella* biotype discrimination. Results of phylogenetic analyses based on polymorphisms of each gene support previous conclusions that *A. tosichella* has two dominant haplotypes in North America [29, 36] with divergence in the ITS1 region similar to the Australian haplotypes EU734729 (WCM1) and EU734726 (WCM2) [29]. Both ITS1 and COI polymorphisms indicated that *A. tosichella* samples obtained in the current study corresponded to lineages MT-1 and MT-8 based on global sampling [48] although ITS sequences generated for the current study were significantly longer (~350bp vs ~617bp). However, several other haplotypes appear to be divergent and may potentially represent additional biotypes [29, 30, 49]. Thus, it was not surprising to determine that the ITS1 and COI polymorphisms determined in our results identified eight new *A. tosichella* haplotypes from wheat with no matches in the GenBank database. These results also confirm previously identified large-scale co-occurrences of biotype 1 and 2 in individual field populations in North America [29, 36, 37], as well as within individual wheat grain heads [50].

Australian types 1 and 2 also co-occur across Australian wheat production areas, although biotype 1 occurs more often in the southeast and biotype 2 more frequently in the west [29, 37]. Results of our experiments further demonstrate possible genetic drift or host shifts in North American *A. tosichella* populations [51], although variation is less than that in observed previously in Turkey [52]. Possible explanations include genetic drift resulting from an invasive *A. tosichella* population [53, 54] or a host shift resulting from *A. tosichella* adaptation to mite resistance genes in wheat [55]. The greater numbers of unique ITS1 region haplotypes in biotype 1 than in biotype 2 (Figs 1, 2 and 3), support conclusions by Harvey et al. [46] that biotype 2 developed after deployment of the *Cmc3 A. tosichella* resistance gene in wheat cultivar TAM 107 in 1983 [56].

Our results also revealed eight haplotypes based on ITS1 region sequence variants, while those of Hein et al. [36] determined only two. The differences in results between the two

studies are likely due to differences in geographic scope of sampling (25 sample sites in our experiments versus 5 sites in Hein et al.) and the year of sample collection (2014–2016 versus 1999). Finally, differences in the results of the two studies may involve an increase in biotype diversity resulting from the recent release of cultivars containing the *Cmc4* gene for *A. tosichella* resistance in Montana and Oklahoma [57, 58], as well as cultivation of cultivars in Colorado, Kansas, Oklahoma and Texas with the *Dn7* gene for resistance the Russian wheat aphid, *Diuraphis noxia* (Kudjumov), and the *H21* gene for resistance to Hessian fly, *Mayetiola destructor* Say, both of which have recently been shown to be resistant to *A. tosichella* [59].

Enders et al. [60] employed a flexible phenological model similar to that used in our experiments to account for spatial or temporal autocorrelation that may have been generated by population dynamics of different cereal aphid virus vectors. Adoption of such models on an area-wide basis in North America could provide an enhanced understanding of *A. tosichella* biotype geographic distribution and improve predictions of the risk of infestations by *A. tosichella* and the viruses they transmit.

Previous reports of the presence of each *A. tosichella* biotype in North America were based on data from mites collected at five locations in one year [36, 61]. In contrast, our experiments are the first to determine *A. tosichella* presence at 38 unique locations from a 1.2 mill km$^2$ area in the North American Great Plains over a 3-year period. As a result, these data provide the first ratios of the two biotypes over multiple locations and years. The use of these data to develop spatial-temporal predictions of *A. tosichella* biotype variation provide the first demonstration of the effects of precipitation and land cover on biotype distribution. The predicted biotype ratios based on 2014, 2015, and 2016 infestations necessitate continued and coordinated monitoring of North American *A. tosichella* biotype variation in order to anticipate future mite infestation intensity and biotype composition.

## Materials and methods

### Sample collection

*Aceria tosichella* was collected from wheat *T. aestivum* heads at 25 locations in the U.S. Great Plains wheat production area from May 21 to July 10, 2014; June 25 to July 12, 2015; and June 11 to June 16, 2016. No special permits were required to collect samples, as verbal permission was given by producers at each sample collection site. The geographic coordinates of each sample location in 2014 and 2015 (S1 Table) or 2016 (S2 Table) were recorded using a hand-held GPS device. In 2014 and 2015, three field sites were sampled at each GPS location and 30 wheat heads were sampled within each of the three field sites, resulting in a total of 90 heads per location. To avoid bias, the heads were grouped, and 10 heads were arbitrarily selected from each 90 head group. One individual live female was transferred using a 30X microscope, from each of the 10 selected wheat heads to a cold microcentrifuge PCR tube and centrifuged at 14,000 rpm at 4˚C for 1 min to position the mite near or in the bottom of the tube before storage at -80˚C. An 8 h recess was observed between transfers to prevent cross-contamination between populations [9]. In 2016, additional collections were made at one location in Kansas, two locations in Missouri, and one location in Nebraska at the same locations as in 2014 and 2015, or in the nearest wheat field in locations where wheat had been rotated (S2 Table). Three wheat fields were sampled at each location and in each field, five heads were sampled from plants at each of three different sites in each field, resulting in a total of 45 heads per location. Each head was kept separate in a plastic bag in order to distinguish genetic differences between mites within a field and a grain head. Mites from each location were previously classified as biotype 1 avirulent to *Cmc3* or biotype 2 virulent to *Cmc3* [47] using methods of Harvey et al. [46].

**Table 3. Primers used to amplify nuclear ribosomal internal transcribed spacer one (ITS1) and cytochrome oxidase I (COI) in *A. tosichella* biotype 1 and 2.**

| Region | Primer name | Sequence | Reference |
|--------|-------------|----------|-----------|
| rDNA—ITS1 | WCM_ITS1_A_F | 5'-GTG AGG CAT CTG GAC TTG CT-3' | This study |
| | WCM_ITS1_A_R | 5'-TTG TTT GCA CGC AGT CAT GG-3' | This study |
| | WCM_ITS1_B_F | 5'-ATC CTT CAT CAC GAC TCG GC-3' | This study |
| | WCM_ITS1_B_R | 5'-CCC TCA TAC AGG CAA GGC TC-3' | This study |
| mtDNA—COI | 1718 F | 5'-GGAGGATTTGGAAATTGATTAGTTCC-3' | [72] |
| | bcdR04 | 5'-TATAAACYTCDGGATGNCCAAAAAA-3' | [73] |

## *Aceria tosichella* DNA processing and amplification

*Aceria tosichella* DNA was extracted using the MyTaq™ Extract-PCR kit (Bioline USA Inc. Taunton, MA). A master mix was prepared for each reaction using 35 μl nuclease-free water (Ambion Co., Lewisville, TX), 10 μl Buffer A and 5 μl Buffer B (total 50 μl). This solution was added to each tube containing a specimen of *A. tosichella*. Tubes were incubated at 75°C and 95°C for 10 min each and thereafter held at 12°C for ∞. Mite DNA extracts were stored at 4°C. Polymerase chain reactions were performed to amplify 618 base pairs (bp) of the nuclear ribosomal internal transcribed spacer 1 (ITS1) region. Primer3Plus [62] was used to design primers to amplify 600 bp of this gene (Table 3). Using primers indicated in Table 3, a subsample of specimens was subjected to cytochrome oxidase I (COI) analysis to confirm whether biotype groupings/designations were correct, and to determine whether analysis of this gene was more adept at detecting variation at local geographic scales. For this reason, the majority of specimens (42 out of a total of 49) for which the COI gene was sequenced were collected in 2016.

All PCRs were conducted in a 40 μl volume including 1 μl DNA extract, 20 μl Taq DNA polymerase (Bioline Inc. Taunton, MA), 0.5 pmol each of the forward and reverse primers (Table 3), 1 μl MgCl$_2$ (Thermo Scientific, New Hampshire, MA), and 17 μl nuclease-free water, using a T100 thermal cycler (Bio-Rad, Hercules, CA). The ITS1 region amplification protocol was 95°C for 3 min (initial denaturation), four cycles of 95°C for 20 sec, 56°C for 15 sec, 72°C for 20 sec, followed by 34 cycles of 95°C for 20 sec, 45°C for 15 sec, 72°C for 20 sec, and 72°C for 15 min. The COI amplification protocol was 95°C for 3 min (initial denaturation), 40 cycles of 95°C for 20 sec, 45°C for 15 sec, 72°C for 20 sec, and 72°C for 15 min. 5 μl of each PCR product was mixed with 1 μl loading dye (Promega, Madison, WI) and run on a 1% agarose gel (Fisher Scientific, Suwanee, GA), stained with GelGreen® Nucleic Acid Gel Stain (Bioline Inc. Taunton, MA) for 60 min and visualized under UV light (Bio-Rad Gel Doc EZ System Gel Imaging System, San Jose, CA) to determine amplification products. PCR product sizes were assessed using the Hi-Lo™ DNA marker (Minnesota Molecular, Inc. Minneapolis, MN) and product concentration was measured by comparison with Lambda DNA of standard concentrations (Promega) and Nanodrop spectrophotometry (Thermo Scientific).

Sanger sequence data were then generated using GeneWiz Inc. (South Plainfield, NJ). Because of large sample sizes, PCR products were sequenced for a few specimens in both directions (F and R) using the same primers used for PCR. However, the majority of our specimens were sequenced in one direction (F) only. Sequences for *A. tosichella* and related species obtained from GenBank were aligned and edited using BioEdit V. 7 software [63].

## Bayesian phylogenetic analyses

Bayesian phylogenetic analyses of the data were performed using MrBayes 3.2 [64]. DNAsP v. 5.10.01 [65] was used to test sequence polymorphism among individuals with ITS1 region and

COI, and ITS1 region sequences were imported as nexus files to POPART [66] to create haplotype network diagrams. The nucleotide sequences of ITS1 region and COI used in phylogenetic analyses have been deposited in GenBank (accession numbers MT336812-MT337241 for ITS1 region, and MT370025-MT370073 for COI).

To delineate between biotype 1 and 2 ITS1 region nucleotide sequences, comparisons were made using GenBank sequence EU734729 serving as a reference for biotype 1 and EU734726 serving as a reference for biotype 2 [29]. For this gene, comparisons were also made using sequences that correspond to lineages MT-1, MT-7 and MT-8 [48]. To delineate between biotype 1 and 2 COI nucleotide sequences, comparisons were made using GenBank sequence JQ248920 serving as a reference for biotype 1. No explicit reference sequence for biotype 2 was available for this gene at the time of the analyses. However, comparisons were also made for this gene using sequences from lineages MT-1, MT-7 and MT-8 [48]. Genbank sequence JF920113 obtained from *Aceria eximia* was used as an outgroup control in the ITS1 sequence analyses, and Genbank sequence FJ387563 obtained from *Aceria* tulipae was used as an outgroup control in the COI sequence analyses. The threshold of sequence similarity required to determine a biotype was 99% identical to a known biotype ITS sequence.

### Spatio-temporal prediction of *A. tosichella* biotype

A generalized additive model [67] was used to capture the spatio-temporal dynamics in the prevalence of *A. tosichella* biotypes 1 and 2, incorporating weather and land cover as dependent variables with temporal changes in *A. tosichella* population dynamics. A binomial distribution was assumed, with the number of "trials" of the binomial distribution being the number of mites sampled at each unique site and time period, which was 10 in 2014–2015 and 15 in 2016.

For each sample obtained, the PRISM database [68] was used to obtain the average monthly temperature and precipitation occurring during the month and the month prior to sample collection, and the 2011 National Land Cover Database [69] was used to determine either grass/pasture or cropland land cover covariates at the 30 m by 30 m resolution. NLCD classes 71 and 82 defined grass/pasture and class 42 defined cropland. Land cover was assumed to influence mite prevalence at a scale larger than 30 m x 30 m resolution. The effective scale influencing the response was determined by calculating the percentage of grass/pasture and cropland within circular regions centered at the sample location with a diameter of 100-, 500-, 1000-, 2500-, 5000-, and 10000m.

Spatio-temporal effects unrelated to weather or land cover covariates i.e., autocorrelation [70] were included using a categorical factor composed of the year of data collection and thin plate regression splines, a type of basis function that models "smooth" effects of spatial location or time [70]. The interaction between grass/pasture and cropland land cover at the 500m scale was included in a given model, but candidate models were constructed for spatial scales at 100-, 500-, 1000-, 2500-, 5000-, and 10000m. The appropriate scale was chosen from the candidate model with the lowest Akaike's information criterion (AIC) score [71] and calculating the AIC. The drivers of the prevalence of each *A. tosichella* biotype were assumed to covariates with coefficients within 90% confidence intervals that did not contain zero.

### Supporting information

**S1 Fig. Probability of *A. tosichella* biotype 1 occurrence (mean ± SE) at locations in Barton (A), Pettis (B), Cape Girardeau (C), Pike (D), Copper (E) and Stoddard (F) county Missouri in 2014, 2015 and 2016.**
(PPTX)

**S2 Fig. Probability of *A. tosichella* biotype 1 occurrence (mean ± SE) at locations in Hughes (A), Lake (B) and Tripp (C) county South Dakota in 2014, 2015 and 2016.**
(PPTX)

**S3 Fig. Probability of *A. tosichella* biotype 1 occurrence (mean ± SE) at locations in Bottineau (A) and Ward (B) county North Dakota in 2014, 2015 and 2016.**
(PPTX)

**S4 Fig. Probability of *A. tosichella* biotype 1 occurrence (mean ± SE) at locations in Cheyenne (A), Hayes (B), Furnas and Saunders (D) county Nebraska in 2014, 2015 and 2016.**
(PPTX)

**S5 Fig. Probability of *A. tosichella* biotype 1 occurrence (mean ± SE) at locations in Barton (A), Ellis (B), Geary (C), Dickinson (D), Greeley (E), Ellsworth (F), Saline (G) and Finney (H) county Kansas in 2014, 2015 and 2016.**
(PPTX)

**S6 Fig. Probability of *A. tosichella* biotype 1 occurrence (mean ± SE) at locations in Dallam (A) and Randall (B) county Texas in 2014, 2015 and 2016.**
(PPTX)

**S1 Table. State, county, and geographic coordinates for locations of *A. tosichella* samples collected in 2014 and 2015.**
(DOCX)

**S2 Table. State, county, and geographic coordinates for locations of *A. tosichella* samples collected in 2016.**
(DOCX)

**S3 Table. Estimates of genetic distance and genetic identity in eight U.S. *A. tosichella* populations collected in 2014, 2015 and 2016, using variation among unique ITS1 haplotypes.**
(DOCX)

**S4 Table. Estimates of genetic distance three U.S. *A. tosichella* populations collected in 2014, 2015 and 2016, using variation among unique COI haplotypes.**
(DOCX)

**S1 Analysis. Reproducible analysis.**
(PDF)

**S1 File.**
(RTF)

## Acknowledgments

Authors gratefully acknowledge the USDA/NIFA North Central IPM Program, the Kansas Wheat Commission, the Kansas Association of Wheat Growers, University of Baghdad and Kansas State University Research and Extension for supporting this research. Contribution no. 20-114-J from the Kansas Agricultural Experiment Station.

## Author Contributions

**Conceptualization:** Luaay Khalaf, Wen-Po Chuang, C. Michael Smith.

**Data curation:** Luaay Khalaf, Alicia Timm, Wen-Po Chuang, Laramy Enders.

**Formal analysis:** Luaay Khalaf, Alicia Timm, Laramy Enders, T. J. Hefley.

**Funding acquisition:** C. Michael Smith.

**Investigation:** Luaay Khalaf, Wen-Po Chuang, Laramy Enders, T. J. Hefley.

**Methodology:** Wen-Po Chuang, Laramy Enders, T. J. Hefley.

**Software:** Alicia Timm, T. J. Hefley.

**Supervision:** C. Michael Smith.

**Writing – original draft:** Luaay Khalaf, T. J. Hefley, C. Michael Smith.

**Writing – review & editing:** Luaay Khalaf, T. J. Hefley, C. Michael Smith.

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
