## [Decision Letter · Decision Letter 0]

10 Feb 2020

PONE-D-19-36077

Spatio-temporal modeling of Aceria tosichella biotype occurrence and distribution

PLOS ONE

Dear Dr. Smith,

Thank you for submitting your manuscript to PLOS ONE. Your manuscript was reviewed by two experts on wheat curl mites and received a split decision: reviewer 1 accepted with minor revision, while reviewer 2 rejected. However, both reviewers raised several important issues to improve the quality of your manuscript. Reviewer 1 suggested several minor but important suggestions. I suggest you consider all of the reviewer 1 suggestions in your revised manuscript. Reviewer 2 is more critical of experimental design in terms of sample collection, analyses, and interpretation of data. I suggest you consider all of the reviewer 2 suggestions into your revised manuscript. Based on reviewers' criticism and on my own assessment of your manuscript, we feel that it has merit but does not fully meet PLOS ONE’s publication criteria as it currently stands. Therefore, we invite you to submit a revised version of the manuscript that addresses the points raised during the review process.

We would appreciate receiving your revised manuscript by Mar 26 2020 11:59PM. To enhance the reproducibility of your results, we recommend that if applicable you deposit your laboratory protocols in protocols.io, where a protocol can be assigned its own identifier (DOI) such that it can be cited independently in the future. For instructions see: http://journals.plos.org/plosone/s/submission-guidelines#loc-laboratory-protocols

We look forward to receiving your revised manuscript.

Kind regards,

Satyanarayana Tatineni, Ph.D

Academic Editor

PLOS ONE

Journal Requirements:

4. We note that Figure 4 in your submission contains satellite images which may be copyrighted. All PLOS content is published under the Creative Commons Attribution License (CC BY 4.0), which means that the manuscript, images, and Supporting Information files will be freely available online, and any third party is permitted to access, download, copy, distribute, and use these materials in any way, even commercially, with proper attribution. For these reasons, we cannot publish previously copyrighted maps or satellite images created using proprietary data, such as Google software (Google Maps, Street View, and Earth). For more information, see our copyright guidelines: http://journals.plos.org/plosone/s/licenses-and-copyright.

You may seek permission from the original copyright holder of Figure 4 to publish the content specifically under the CC BY 4.0 license. 

If you are unable to obtain permission from the original copyright holder to publish these figures under the CC BY 4.0 license or if the copyright holder’s requirements are incompatible with the CC BY 4.0 license, please either i) remove the figure or ii) supply a replacement figure that complies with the CC BY 4.0 license. Please check copyright information on all replacement figures and update the figure caption with source information. If applicable, please specify in the figure caption text when a figure is similar but not identical to the original image and is therefore for illustrative purposes only.

5. Please include your tables as part of your main manuscript and remove the individual files. Please note that supplementary tables (should remain/ be uploaded) as separate "supporting information" files.

Reviewers' comments:

Reviewer's Responses to Questions

**Comments to the Author**

1. Is the manuscript technically sound, and do the data support the conclusions?

Reviewer #1: Yes

Reviewer #2: No

2. Has the statistical analysis been performed appropriately and rigorously? 

Reviewer #1: I Don't Know

Reviewer #2: No

3. Have the authors made all data underlying the findings in their manuscript fully available?

Reviewer #1: Yes

Reviewer #2: Yes

4. Is the manuscript presented in an intelligible fashion and written in standard English?

Reviewer #1: Yes

Reviewer #2: No

5. Review Comments to the Author

Reviewer #1: The manuscript by Khalaf et al. addresses the regional distribution of two primary genetically distinct types (Type 1 and Type 2) of the wheat curl mite across a major portion of the Great Plains. Thirty-seven locations across six states were sampled over three years and the occurrence of each of these two distinct genotypes was determined. This study provides an important delineation of the distribution of these two mite types and also provides evidence of additional genetic variation within these two types. This information provides an important contribution to our understanding of the wheat curl mite and the distribution of the genetic diversity of the mite. The distribution model developed is interesting but a less important part of the paper. While there was a lot of work that went into this paper, the model needs to be validated with independent data to determine its accuracy. The manuscript is an important contribution and should be published, but it does require some modifications to address the following points: (Many of these are not major revisions but they need to be addressed.)

- First, I do not think that the use of the term ‘biotypes’ in the title is an appropriate term to describe these types. The term biotype has often taken on a very generic meaning and as such would be appropriate in this sense. However, the authors refer to important differences between these two types in terms of host plant resistance and reference the importance of mite resistance in wheat. In this (HPR) regard, ‘biotype’ is used to separate groups that ‘differ in their ability to utilize a particular trait in a plant genotype’ (Smith 2005) and these are detected by use of differential host genotypes. Studies undertaken by Harvey et al. 1999 and reviewed in Smith (2005) show that the populations tested from MT have a clearly distinct reaction from other Type 1 groups to the differentials provided, but the work by Hein et al. 2012 shows that the MT population is included with other groups of Type 1 mites (0% and 0.5% nucleotide differences from other Type 1 groups for ITS1 and COI, respectively). Thus, within the Type 1 mites there are at least 2 distinct ‘biotypes’ (as defined above for HPR relationships). The authors do raise this possibility in lines 184-85 as being a potential, but this previous information indicates that it is more than potential. Thus, if there are more than one ‘biotype’ within the Type 1 grouping should Type 1 and Type 2 mites be referred to as ‘biotypes’?

- L. 34-35 and l. 73 – Authors state that mite resistant cultivars are the ‘only effective method of controlling’ the mite. It is agreed that this resistance would be a valuable tool for management, but cultural practices, primarily controlling over-summering hosts (e.g. volunteer wheat), when applied properly can be very effective and are primarily relied on currently for management of this mite-virus complex.

- L. 81-84 – In addition to the points listed, the most dramatic difference between Type 1 and Type 2 mites is the ability of Type 2 and inability of Type 1 mites to transmit Triticum mosaic virus (see McMechan et al 2014).

- L. 108-110 – The statement that the ranges of genetic distance between ITS1 and COI are ‘similar’ is not accurate as the upper value of the range for COI is over 7 times higher than for ITS1 (0.204 vs 0.028).

- Results in lines 125-129 do not match clearly to the Figures (2, 3) and Fig 3 has a mismatched label and caption (Barton field 3 vs 1?).

- L. 139-145 – In the discussion of biotype ratios, reference is made to significant changes in ratios but no statistics or statistical methods are presented on how this was determined. I think it would be quite valuable to include Tables S5 and S6 to give the reader an indication of how variable these ratios are. These tables are actually more valuable than Figures 1-3 for understanding what is going on with the data. For example, in 2016 at Cape Girardeau, the average ratio is 8:2 but they range from 10:0 to 0:10. From this it seems that mention of slight changes in these ratios are probably not that important. One important point that is not emphasized is that these rations often change greatly even within a field.

- Lines 146 + - These results present the general results of the modeling but there are no statistics presented as to how well the model fits the data. ‘Negative coefficient estimate’ are mentioned but none are given. There needs to be some determination as to how well the model fits the data rather than just saying the ‘confidence interval did not include zero’. Also in the discussion there needs to be some acknowledgement that this model was developed using the data generated but there has been no validation of this model with independently collected data to provide some confidence of its true accuracy.

- L. 216 – the temperature values provided from Kuczynski et al 2016 are reversed as Type 1 is 32C and Type 2 is 35C.

- L. 216-220 and 239-242 – Through the paper (abstract and other areas) there is mention of the model being ‘temporal’ but the temporal factor is limited (this is noted partly here) by limited sampling primarily to June (e.g. little done in May, some states only sampled in one month, and sampling in 2016 only done over 6 days in June). It seems that extension of the model and mapping in Fig 4 to cover May-June-July is a bit of a stretch outside of the inference space of the database it was developed on. It is fine to make predictions with the model, but it should be made clear in the discussion that this is the case and also emphasize the need for further validation.

- L. 255-261 – The description of the sampling in 2016 is not described very completely. ‘45 heads collected per location’ but only 9 selected (I think from Table S6) but how? And how many mites were taken from a head for assay? 10 (again from Table S6?

- L. 270 – A ‘subset’ of specimens taken for COI – how big a subset and how selected?

- L. 310-312 – ‘No explicit reference sequence for biotype 2 was available’, but Genbank sequence for Type 2 are available from published reports. Skoracka et al. 2014 list several COI sequences from different studies and different locations. According to Skoracka et al 2014 the JQ248920 sequence from line 311 is identified as MT-1 which would be Type 2 in this study – not Type 1 as indicated here.

Reviewer #2: The manuscript led by C. M. Smith raises a relevant issue concerning the occurrence and distribution of an important phytophagous mite species (Aceria tosichella), a global pest of cereals. Accurate predicting species distributions and explaining which environmental factors influence distribution is a fundamental goal of ecology and, in the case of pests and parasites, is an integral part of practical applications such as management strategies. For A. tosichella such affairs are especially hindered due to the evidence that this minute mite actually represents a species complex consisting of at least 29 of different biotypes (genetic lineages) which are morphologically undistinguishable, but may differ in biology and ecology. The authors of this manuscript aim to assess spatial and temporal changes in the occurrence of two mite biotypes and to build a model allowing for the prediction of the probability of biotypes occurrence at unexamined locations and dates. They also intend to find out which environmental variables explain A. tosichella biotypes distribution, and they believed that their results will improve predictions of the risk of mites infestations. All these goals are ambitious and worthwhile, but unfortunately I regret to grade the presented study as unsound and not making a valid contribution to the scientific record.

There are several weaknesses, among which the more important are flaws in experimental set-up and resulting unjustified conclusions based on the data. To investigate spatio-temporal changes in organisms’ distribution a rigorous experiment with appropriate sampling scheme, replication and sample size is required. Sampling scheme should achieve an even distribution of sampling localities, and sampling localities should be randomized. The same localities should be replicated in each year through the whole study period to achieve adequate resolution of temporal variation. None of these requirements are met in the study presenting by the authors. For example, sampling in 2016 was done only during 3 days in June, whereas in 2014-2015 in May, June, July. There is no map provided the visual information of sampling localities which informing about even (or not even) distribution of localities. There is a lack of information how the sampling localities have been selected. Finally, the total number of sampled localities (less than 40) from a 1.2 mil km2 area is definitely too low to justify the correctness of modeling the distribution, thus to produce robust results and draw sound conclusions.

The description of the modeling in Methods (‘Spatio-temporal prediction of A. tosichella biotype’) is laconic and sketchy, what makes it impossible to assess the correctness of this analysis (which, by the way, is based on inaccurate data sampling). The presentation of distribution results is very strange and incomprehensible. Fig 4: the probability of biotypes occurrence marked by different colors - blue for biotype 2 and red for biotype1. But the biotypes can co-occur (what is known from both previous studies and the study conducted by the authors of this manuscript). So how to find out the probability of their co-occurrence on this figure? The models should be presented separately for each biotype. The supplementary figures present the probability occurrence for each locality and year and month separately, what is uninformative when the goal is to detect any patterns in spatial and temporal distribution.

Another thing about which I am concerned there is a lack of information in ethics statement about the permission regarding field study (‘N/A’ according to the authors and no information in M&M section). Cereals field form which the authors collected wheat samples are most likely to be state-owned or privately owned (unless they are authors-owned). As such permits and approvals obtained for the work, including the full name of the authority that approved the study is required.

Another major drawback of this manuscript there is incomplete and outdated literature cited by the authors, which may introduce disinformation about the very important economically mite species, what may impair both the basic and applicative science. The authors ignore recent articles that are strongly related to the issues they address as well as some review articles summarizing latest evidences about Aceria tosichella biology and ecology. For example, the article in Plos One is about spatial distribution of A. tosichella biotypes, and in the article from BMC Evolutionary Ecology the genetic structure and haplotypes networks of A. tosichella biotypes are discussed.

References e.g. 28, 29, 39 are hardly to access and in fact are not adequately representative to be cited in the context presented by the authors.

(Skoracka A., Lewandowski M., Rector B.G., Szydło W., Kuczyński L. 2017. Spatial and Host-Related Variation in Prevalence and Population Density of Wheat Curl Mite (Aceria tosichella) Cryptic Genotypes in Agricultural Landscapes. PLoS ONE 12(1): e0169874. DOI: 10.1371/journal.pone.0169874

Skoracka A., Lopes L. F., Alves M. J., Miller A., Lewandowski M., Szydło W., Majer A., Różańska E. i Kuczyński L. 2018. Genetics of lineage diversification and the evolution of host usage in the economically important wheat curl mite, Aceria tosichella Keifer, 1969. BMC Evolutionary Biology 18: 122, https://doi.org/10.1186/s12862-018-1234-x

Singh K., Wegulo S. N., Skoracka A., Kundu J. K. 2018. Wheat streak mosaic virus: a century old virus with rising importance worldwide. Molecular Plant Pathology 19(9): 2193-2206, https://doi.org/10.1111/mpp.12683

Skoracka A., Rector B. G., Hein G., L. 2018. The interface between wheat and the wheat curl mite, Aceria tosichella, the primary vector of globally important viral diseases. Frontiers in Plant Science 9 (1098), 1-8, https://doi.org/10.3389/fpls.2018.01098)

Finally, it is not clear why the authors focus on phylogenetic analyses (in M&M and results sections). To fulfill their aims, the authors would simply need to discriminate the biotypes on the basis of DNA barcodes using blast function and estimate genetic distances. There is no need to employ any phylogenetic analysis, the more than the authors do not specify any hypothesis to which testing they would need to apply e.g. Bayesian Inference.

The presented haplotypes network are uninformative. What kind of scientifically sound information emerges from the picture that about 50% of a given haplotype was present in Kansas or on head 1 of one of the three inspecting wheat heads? Why the ratio of biotypes is important? What scientific conclusions we could draw from these results? If we are interested in biotype 1 & 2 occurrence, it would be more explanatory to see the map visualizing the proportion of biotypes in a given locations.

The identification of biotypes is under a question (M&M line 312). I am sure that in GenBank there is more than one reference sequence for biotype 2. Personally I have submitted a plenty of them and I am sure that other experts studying A. tosichella submitted the COI sequences, too.

The discussion is also a very weak point on this manuscript. Some sentences and conclusions are trying to be supported by the literature that is out (or at least very far) of the subject (e.g. lines 192-195). No sound conclusions supported by the data and results arise.

Except of these major concerns above, below are listed some other remarks regarding the abstract and introduction.

Title: ‘biotypes’ instead of ‘biotype’ should be used since the study is based on two A. tosichella biotypes

Abstract: lines 35-36: something is missing in this sentence.

Lines 36-38: Previous studies have detected many more biotypes than just two.

Line 39: please use the full genus name when start the sentence from Aceria tosichella. Please check and correct the whole text considering this remark (e.g. lines 59, 70) and also please check if the Latin mite name is consequently written in italics (e.g. line 50).

Lines 48-51: This conclusions suggests that biotypes 1 and 2 differ in their response to mite-resistant wheat varieties. But the study is in fact not about this, but on the biotypes occurrence and distribution. So, data presented in the manuscript did not support this conclusion.

Key words: Cmc – is not clear

Line 70-71: ‘cryptic behavior’: what do you mean?

Line 71: There are plenty of recent research that fit to be quoted here. I suggest to cite Navia et al. 2013 (https://link.springer.com/article/10.1007/s10493-012-9633-y) instead of 28 and 29. These two references are hardly to access and in fact are not adequate representative. Moreover, from several years there is an evidence that wheat curl mite is a complex of species consisting of biotypes with divergent host specificity, ranging from specialists to generalist. This should be underlined as this situation also significantly impair the detection and research on A. tosichella.

Line 76: [39] this reference is also hardly to access. Can authors here quote here to published article?

Line 79: the ‘biotype’ occurs here for the first time in the Introduction, thus the phrase ‘each biotype’ is not clear here. Authors should first introduce the readers to the issue of many biotypes identified within wheat curl mite. There is important that not only two biotypes were identified but seven in the articles that authors cite (42, 43). Further research discovered 16 and 29 biotypes respectively: https://doi.org/10.1371/journal.pone.0169874 ; https://bmcevolbiol.biomedcentral.com/articles/10.1186/s12862-018-1234-x). This issue needs to be clarified, because in the present form the background of this study lacks very important information about biology and ecology of A. tosichella and misleadingly suggests readers that only two biotypes exists. Also, authors mention (lines 80-81) that biotypes identification was made on the basis on ITS1 and COI, but ITS2 and 28S rDNA D2 regions were also used (references: 42,43).

Line 83: these references (46-48) refer to Australia and North America only, not to Europe and South America as authors state: “biotypes co-occur in mixed populations within each continent.” The reference to Europe could be: https://doi.org/10.1371/journal.pone.0169874.g002 (biotype 1 corresponds to MT-1, and biotype 2 to MT-1; you can read about this in the recent reviews (Skoracka et al. 2018a; Singh et al. 2018).

Line 83: Change to: “At least 29 genetic lineages exist in A. tosichella complex in the world [Skoracka et al. 2018b: https://doi.org/10.1186/s12862-018-1234-x].”

Line 85: biotype 2 is mentioned in the Introduction here for the first time. Again there is necessary to develop the issue of wheat curl mite biotypes before the formulation of the objective of this study.

Line 91: change ‘structure’ to ‘dynamics’

6. PLOS authors have the option to publish the peer review history of their article (what does this mean?). If published, this will include your full peer review and any attached files.

Reviewer #1: No

Reviewer #2: No

---

## [Author Response · Author response to Decision Letter 0]

1 May 2020

Reviewer #1: The manuscript by Khalaf et al. addresses the regional distribution of two primary genetically distinct types (Type 1 and Type 2) of the wheat curl mite across a major portion of the Great Plains. Thirty-seven locations across six states were sampled over three years and the occurrence of each of these two distinct genotypes was determined. This study provides an important delineation of the distribution of these two mite types and also provides evidence of additional genetic variation within these two types. This information provides an important contribution to our understanding of the wheat curl mite and the distribution of the genetic diversity of the mite. The distribution model developed is interesting but a less important part of the paper. While there was a lot of work that went into this paper, the model needs to be validated with independent data to determine its accuracy. 

Author Response: We thank reviewer #1 for their insightful and careful review. We have addressed all of reviewer #1’s comments in our responses that follow and many of the suggestions have improved the quality and clarity of our work. In the comment above, it appears that the main concern about our work is that we did not validate our spatio-temporal statistical model with independently collected data. Due to the complexity of our research, the first author decided to collaborate with a professional statistician (Dr. Trevor Hefley) who is a faculty member in the Department of Statistics at Kansas State University. In consultation with our research group, the entire statistical analysis was conducted by Dr. Hefley. The carefulness of Dr. Hefley’s statistical analysis is demonstrated in the supporting material which contains extensive detail and computer code required to reproduce all results reported in our manuscript (see Reproducible Analysis pdf in the supporting material).

While we do agree with the notion that all statistical models should be validated with independent data, doing so is in direct conflict with how science is currently funded or conducted. To do what the reviewer is suggesting, we would need to go collect a second data source. Prior to collection of this second data source, we would use the data from our study to make spatio-temporal predictions (i.e., essentially the predictive heatmap in Fig. 4 of our paper). We would then collect the second data source and report how the predictions matched the new independent data source. While we do agree that this process is the gold standard to validate statistical models, it is something achieved by very few (if any) studies. Since very few studies can justify collecting a second data set that is used only for model validation, there are many alternative procedures that attempt to do the impossible and assess predictive accuracy without an independent data set. Such procedures have a long history in applied statistics and include cross-validation and model selection using information criterion. All of these procedures are simply not as good as the gold standard and can easily be “hacked,” but are widely used because there is a need to assess predictive accuracy in the absence of a truly independent data set.

In our original work, we used Akaike information criterion (AIC) to perform model selection. The AIC was initially derived to “find” the most accurate predictive model from a set of candidate models. In other words, AIC was derived to “estimate” the accuracy of predictions from a statistical model (relative to other models) that would be obtained from a second (independent) data source. Of course, nothing is free of flaws and AIC has its own and extra assumptions, but for our analysis it is likely the best that we can do. We have added clarification to lines 229 and 336 that states that AIC was used to measure the relative predictive accuracy of our models and that future studies may want to consider collecting a second independent data source.

Reviewer #1: The manuscript is an important contribution and should be published, but it does require some modifications to address the following points: (Many of these are not major revisions but they need to be addressed.)

- First, I do not think that the use of the term ‘biotypes’ in the title is an appropriate term to describe these types. The term biotype has often taken on a very generic meaning and as such would be appropriate in this sense. However, the authors refer to important differences between these two types in terms of host plant resistance and reference the importance of mite resistance in wheat. 

In this (HPR) regard, ‘biotype’ is used to separate groups that ‘differ in their ability to utilize a particular trait in a plant genotype’ (Smith 2005) and these are detected by use of differential host genotypes. 

Studies undertaken by Harvey et al. 1999 and reviewed in Smith (2005) show that the populations tested from MT have a clearly distinct reaction from other Type 1 groups to the differentials provided, but the work by Hein et al. 2012 shows that the MT population is included with other groups of Type 1 mites (0% and 0.5% nucleotide differences from other Type 1 groups for ITS1 and COI, respectively). 

Thus, within the Type 1 mites there are at least 2 distinct ‘biotypes’ (as defined above for HPR relationships). 

Reviewer #1: The authors do raise this possibility in lines 184-85 as being a potential, but this previous information indicates that it is more than potential. Thus, if there are more than one ‘biotype’ within the Type 1 grouping should Type 1 and Type 2 mites be referred to as ‘biotypes’?

Author Response: We agree that because ‘biotypes’ are generally defined based on HPR relationships and not specific nucleotide differences or a threshold of genetic similarity, it is possible that multiple phenotypic ‘biotypes’ could exist within the current Type 1 and Type 2 molecular groupings. We were unable to confirm whether the new haplotypes reported in this study are indeed different biotypes (i.e. no additional screening w/ different host plant genotypes). To address the reviewer’s point we have added additional text explaining the limitations of this study, including the assumption that a 99% sequence similarity threshold (ITS) correlates to true phenotypic biotype variation and lack of further identification of all potential distinct biotypes based on HPR bioassays. see Discussion lines 186-199, and Methods lines 300-311. 

Reviewer #1: - L. 34-35 and l. 73 – Authors state that mite resistant cultivars are the ‘only effective method of controlling’ the mite. It is agreed that this resistance would be a valuable tool for management, but cultural practices, primarily controlling over-summering hosts (e.g. volunteer wheat), when applied properly can be very effective and are primarily relied on currently for management of this mite-virus complex.

Author Response: New text describing cultural practices in the revision cites a reference that has been added to the reference list. 

Reviewer #1: - L. 81-84 – In addition to the points listed, the most dramatic difference between Type 1 and Type 2 mites is the ability of Type 2 and inability of Type 1 mites to transmit Triticum mosaic virus (see McMechan et al 2014).

Author Response: Revised text referring to transmission differences and the McMechan et al 2014 reference have added to the revision. 

Reviewer #1: - L. 108-110 – The statement that the ranges of genetic distance between ITS1 and COI are ‘similar’ is not accurate as the upper value of the range for COI is over 7 times higher than for ITS1 (0.204 vs 0.028).

Author Response: This sentence has been corrected in the phylogenetic analyses subsection of the revised results. 

Reviewer #1: - Results in lines 125-129 do not match clearly to the Figures (2, 3) and Fig 3 has a mismatched label and caption (Barton field 3 vs 1?).

Author Response: Fig. 2A corrected to Barton field 1, Figure 3A legend and caption corrected to Cape Girardeau Field 1. 

Reviewer #1: - L. 139-145 – In the discussion of biotype ratios, reference is made to significant changes in ratios but no statistics or statistical methods are presented on how this was determined. I think it would be quite valuable to include Tables S5 and S6 to give the reader an indication of how variable these ratios are. These tables are actually more valuable than Figures 1-3 for understanding what is going on with the data. For example, in 2016 at Cape Girardeau, the average ratio is 8:2 but they range from 10:0 to 0:10. From this it seems that mention of slight changes in these ratios are probably not that important. One important point that is not emphasized is that these rations often change greatly even within a field.

Author Response: S5 and S6 Tables are included as Tables 2 and 3 in the revision and text added on lines 122 to line 128 of the revision. 

 Reviewer #1: - Lines 146 + - These results present the general results of the modeling but there are no statistics presented as to how well the model fits the data. ‘Negative coefficient estimate’ are mentioned but none are given. There needs to be some determination as to how well the model fits the data rather than just saying the ‘confidence interval did not include zero’. Also in the discussion there needs to be some acknowledgement that this model was developed using the data generated but there has been no validation of this model with independently collected data to provide some confidence of its true accuracy.

Author Response: On line 154 we have added that our selected model has a “deviance explained” of 71%. The deviance explained is similar to the coefficient of determination (R2) that is familiar to most scientists. The R2, however, is appropriate for only linear regression whereas the deviance explained is appropriate for our models. Similar to R2, a value of 100% means a perfect fit of the model to the data (which was the same was used to fit the model) whereas a value of 0% means our model is no more accurate than an intercept-only model. In the revision, we have also conducted and included standard model assumption checking procedures in the supporting material (see Reproducible Analysis pdf file). We have mentioned these procedures on line 229.

The regression coefficients were provided in the supporting material of the original draft (see Reproducible Analysis pdf file). As noted by reviewer #1: “The distribution model developed is interesting but a less important part of the paper.” We agree with this statement and that is why we chose not to use manuscript space to present these exact numerical results. To really do these numbers justice, we would need to report the coefficient estimates and confidence intervals which would require an extra table. Again, these values are in the supporting material.

Finally, and as stated in our response to the first comment by reviewer #1, we have done our best to assess the accuracy of our model. While assessment of accuracy using an independent data is the gold standard, few, if any, studies do this. Lastly, even if we had access to an independently collected data set to assess the accuracy of our model, we disagree with the reviewer that this would enable us to assess the “true” accuracy of our model. The independently collected data set would enable us to estimate the predictive accuracy of our model, but this estimate would contain uncertainty unless our independently collected data set contained all possible data points (across space and time) which would require an infinite amount of sampling for our study. 

Reviewer #1: - L. 216 – the temperature values provided from Kuczynski et al 2016 are reversed as Type 1 is 32C and Type 2 is 35C. 

Author Response: We refrain from interpreting the impact of temperature on the occurrence of biotypes. This reference has been removed. 

Reviewer #1: - L. 216-220 and 239-242 – Through the paper (abstract and other areas) there is mention of the model being ‘temporal’ but the temporal factor is limited (this is noted partly here) by limited sampling primarily to June (e.g. little done in May, some states only sampled in one month, and sampling in 2016 only done over 6 days in June). It seems that extension of the model and mapping in Fig 4 to cover May-June-July is a bit of a stretch outside of the inference space of the database it was developed on. It is fine to make predictions with the model, but it should be made clear in the discussion that this is the case and also emphasize the need for further validation.

Author Response: The reviewer is correct. We have added this warning to the text and to the Fig. 4 legend text.

Reviewer #1: - L. 255-261 – The description of the sampling in 2016 is not described very completely. ‘45 heads collected per location’ but only 9 selected (I think from Table S6) but how? And how many mites were taken from a head for assay? 10 (again from Table S6?

Author Response: The S6 Table title was incorrect. S6 Table is now Table 1 of the revision, with a corrected title shown as “Ratios of A. tosichella biotype 1 and 2 at one location in Kansas, two locations in Missouri, and one location in Nebraska in 2016. A total of three fields at each location were sampled, five individuals were collected at each of three sites in each field for a total of 45 individuals per field.” This text matches the text in the last two sentences of the methods subsection on Sample Collection. 

Reviewer #1: - L. 270 – A ‘subset’ of specimens taken for COI – how big a subset and how selected?

Author Response: Further detail has been added in lines 241 to 244 of the revision. 

Reviewer #1: - L. 310-312 – ‘No explicit reference sequence for biotype 2 was available’, but Genbank sequence for Type 2 are available from published reports. Skoracka et al. 2014 list several COI sequences from different studies and different locations. According to Skoracka et al 2014 the JQ248920 sequence from line 311 is identified as MT-1 which would be Type 2 in this study – not Type 1 as indicated here.

Author Response: Reference to Skoracka et al. 2014 and text regarding the MT-1 and MT-8 sequences in Skoracka et al. 2014 have been added in lines 115- 117 and lines 300-311. 

Reviewer #2: The manuscript led by C. M. Smith raises a relevant issue concerning the occurrence and distribution of an important phytophagous mite species (Aceria tosichella), a global pest of cereals. 

Accurate predicting species distributions and explaining which environmental factors influence distribution is a fundamental goal of ecology and, in the case of pests and parasites, is an integral part of practical applications such as management strategies. For A. tosichella such affairs are especially hindered due to the evidence that this minute mite actually represents a species complex consisting of at least 29 of different biotypes (genetic lineages) which are morphologically undistinguishable, but may differ in biology and ecology. The authors of this manuscript aim to assess spatial and temporal changes in the occurrence of two mite biotypes and to build a model allowing for the prediction of the probability of biotypes occurrence at unexamined locations and dates. They also intend to find out which environmental variables explain A. tosichella biotypes distribution, and they believed that their results will improve predictions of the risk of mites infestations. All these goals are ambitious and worthwhile, but unfortunately I regret to grade the presented study as unsound and not making a valid contribution to the scientific record.

Reviewer #2: There are several weaknesses, among which the more important are flaws in experimental set-up and resulting unjustified conclusions based on the data. To investigate spatio-temporal changes in organisms’ distribution a rigorous experiment with appropriate sampling scheme, replication and sample size is required. 

Author Response: The reviewer seems to be referring to standardized/randomized sampling efforts found in other studies that attempt to estimate abundance or population density (e.g. Skoracka et al. 2017 PlosONE), which is not the objective of the current study – here we use a generalized additive model (GAM) to capture the spatio-temporal dynamics in the probability of occurrence of A. tosichella biotypes 1 and 2. The sample size and distribution of sampling locations used in this study are therefore adequate to achieve resolution of temporal variation in the probability of occurrence (not abundance) of biotypes and meet criteria considered best-practices for design of spatial-temporal data collection. 

Due to the complexity of our research, the first author decided to collaborate with a professional statistician (Dr. Trevor Hefley) who is a faculty member in the Department of Statistics at Kansas State University. While we do respect reviewer #2’s comments, many of the suggestions about the experimental design and statistical analyses, in this and other comments, are simply wrong. Further, reviewer #2 makes suggested changes that are not possible which highlights the reviewer lack of knowledge about statistics and proper experimental design for spatio-temporal data collection. 

In the above comment Reviewer #2 suggests that “To investigate spatio-temporal changes in organisms’ distribution a rigorous experiment with appropriate sampling scheme, replication and sample size is required.” This statement is vague and not true. For example, there are thousands of studies published that rigorously investigate changes in organism’s distribution that have no replication (e.g., see Elith & Leathwick 2009 for an introduction and review). Furthermore, the historical and current philosophy in spatio-temporal statistics is that data collection cannot be replicated unless you can somehow take multiple measurements at the exact same location and time (see Ch. 1 in Cressie and Wikle 2011). For many studies like ours, replication may be conceptually possible, but will provide no information about the change in spatio-temporal distribution. For example, in our study Aceria tosichella were collected from wheat heads from a given field. For our study, the idea of replication means that we would have to resample a given field at the exact same time. Clearly this doesn’t provide information about change (in anything) over time or space since the data were collected at the same time and location. 

Finally, the reviewer suggests that an “appropriate sampling scheme” is required, but provides no guidance on what they think is appropriate. Again, we defer to our co-author that is a professional statistician. Dr. Trevor Hefley doesn’t see any evidence that the sampling scheme is inappropriate and believes that what the authors have done is standard practice for collection of spatio-temporal data.

Elith, J., & Leathwick, J. R. (2009). Species distribution models: ecological explanation and prediction across space and time. Annual review of ecology, evolution, and systematics, 40, 677-697.

Cressie, N., & Wikle, C. K. (2011). Statistics for Spatio-temporal data. John Wiley & Sons.

Reviewer #2: Sampling scheme should achieve an even distribution of sampling localities, and sampling localities should be randomized. The same localities should be replicated in each year through the whole study period to achieve adequate resolution of temporal variation. None of these requirements are met in the study presenting by the authors. For example, sampling in 2016 was done only during 3 days in June, whereas in 2014-2015 in May, June, July. 

Author Response: The spatial sampling design suggested by the review (i.e., one that achieve an even distribution of sampling localities) is just one of many common designs. There is no one right way to sample spatio-temporal data and the reviewer may want to take a look at Mateu and Müller (2012). In order to recommend or declare that a certain sampling designed is to be preferred, one has to clearly state the criteria they wish to optimize. For example, some studies may wish to estimate regression coefficients with the highest level of accuracy and therefore would want to use a targeted design that places sample points in areas where rapid changes (in the prevalence of biotypes) is expected to occur. The same types of arguments hold for time. 

Mateu, J., & Müller, W. G. (Eds.). (2012). Spatio-temporal design: Advances in efficient data acquisition. John Wiley & Sons.

Reviewer #2: There is no map provided the visual information of sampling localities which informing about even (or not even) distribution of localities. 

Author Response: S Tables 1 and 2 show exact GPS coordinates for all locations of sample collection in 2014, 2015 and 2016. 

Reviewer #2: There is a lack of information how the sampling localities have been selected. 

Author Response: See lines 246-260. 

Reviewer #2: Finally, the total number of sampled localities (less than 40) from a 1.2 mil km2 area is definitely too low to justify the correctness of modeling the distribution, thus to produce robust results and draw sound conclusions.

Author Response: The reviewer must justify and define the arbitrary comment that “the total number of sampled localities….. is definitely too low” before the authors can respond. 

Reviewer #2: The description of the modeling in Methods (‘Spatio-temporal prediction of A. tosichella biotype’) is laconic and sketchy, what makes it impossible to assess the correctness of this analysis (which, by the way, is based on inaccurate data sampling). 

Author Response: This comment is disrespectful. “Laconic and sketchy” are vague terms that the reviewer fails to justify with any concrete suggestions as to what is missing from the description of modeling in the methods. Our research group worked with a professional statistician who is a faculty member in the Department of Statistics at Kansas State University. In consultation with our research group, the entire statistical analysis was conducted by Dr. Hefley. A more in-depth description of the methods is provided in the supporting material (see Reproducible Analysis). Similarly, the reviewer provides no justification of claims that data sampling was “inaccurate”. If the reviewer provided more constructive forms of criticism, then we could address them. As stated, these comments are simply unhelpful, unprofessional and do not warrant a response. In the future, we ask the that the reviewer provide constructive comments that we can objectively address rather than vague non-expert opinion. 

Reviewer #2: The presentation of distribution results is very strange and incomprehensible. Fig 4: the probability of biotypes occurrence marked by different colors - blue for biotype 2 and red for biotype1. But the biotypes can co-occur (what is known from both previous studies and the study conducted by the authors of this manuscript). So how to find out the probability of their co-occurrence on this figure? 

Author Response: To determine the probability of co-occurrence you can use elementary probability. For example, if the probability of biotype 1 is 0.25 at a location then the probability of biotype 2 has to be 0.75 (i.e., 1 – 0.25 = 0.75). The probability of co-occurrence is 0.25 x 0.75 = 0.1875. 

Reviewer #2: The models should be presented separately for each biotype. The supplementary figures present the probability occurrence for each locality and year and month separately, what is uninformative when the goal is to detect any patterns in spatial and temporal distribution.

Author Response: The data we have is aggregated binary data (i.e., the sum of zero’s and ones). Each mite is either biotype 1 or 2. Binary data codes this as y=1 for biotype 1 and y=2 for biotype 2. If we preform the analysis on each biotype separately, we then have “presence-only” data. For example, we would have to conduct an analysis of biotype 1 using only the observations where y = 1. At this point there is no variability in the response variable (because all y’s are equal to one). There are statistical methods that can be used for presence-only data, but the use for our data is nonsensical due to a contrived separation of the binary data into two “presence-only” data sets.

Reviewer #2: Another thing about which I am concerned there is a lack of information in ethics statement about the permission regarding field study (‘N/A’ according to the authors and no information in M&M section). Cereals field form which the authors collected wheat samples are most likely to be state-owned or privately owned (unless they are authors-owned). As such permits and approvals obtained for the work, including the full name of the authority that approved the study is required. 

Author Response: No special permits were required (see revised text in the Materials and Methods section). Verbal permission was given by growers at each sample collection site.

Reviewer #2: Another major drawback of this manuscript there is incomplete and outdated literature cited by the authors, which may introduce disinformation about the very important economically mite species, what may impair both the basic and applicative science. The authors ignore recent articles that are strongly related to the issues they address as well as some review articles summarizing latest evidences about Aceria tosichella biology and ecology. For example, the article in PLoSONE is about spatial distribution of A. tosichella biotypes, and in the article from BMC Evolutionary Ecology the genetic structure and haplotypes networks of A. tosichella biotypes are discussed.

Author Response: The two suggested articles (Skoracka et al. 2017 and 2018) focus on host plant preference (domesticated & wild grasses) and distribution of different genetic lineages found within the WCM species complex and are only slightly relevant to the current manuscript. However, we have referenced the 2017 PLoSONE article that included genetic lineages corresponding to biotypes 1 and 2 as described at the beginning of the discussion. 

Reviewer #2: References e.g. 28, 29, 39 are hard to access and in fact are not adequately representative to be cited in the context presented by the authors.

Author Response: Citations 28 and 29 have been replaced with [Skoracka et al. 2018. Frontiers in Plant Science 9:1098, and Skoracka et al. 2018.BMC Evolutionary Biology 18:122], citation 39 has been removed.

Reviewer #2: Finally, it is not clear why the authors focus on phylogenetic analyses (in M&M and results sections). To fulfill their aims, the authors would simply need to discriminate the biotypes on the basis of DNA barcodes using blast function and estimate genetic distances. There is no need to employ any phylogenetic analysis, the more than the authors do not specify any hypothesis to which testing they would need to apply e.g. Bayesian Inference.

Author Response: The phylogenetic analyses have been removed.

Reviewer #2: The presented haplotypes network are uninformative. What kind of scientifically sound information emerges from the picture that about 50% of a given haplotype was present in Kansas or on head 1 of one of the three inspecting wheat heads? Why the ratio of biotypes is important? What scientific conclusions we could draw from these results? If we are interested in biotype 1 & 2 occurrence, it would be more explanatory to see the map visualizing the proportion of biotypes in a given locations.

Author Response: Haplotype networks are included to provide a visual representation of biotype distributions at various spatial scales. They show that biotype occurrence is not constant at most spatial levels. While a map visualizing biotype proportions would be useful, it would also be impractical given the scale of sampling. 

Reviewer #2: The identification of biotypes is under a question (M&M line 312). I am sure that in GenBank there is more than one reference sequence for biotype 2. Personally I have submitted a plenty of them and I am sure that other experts studying A. tosichella submitted the COI sequences, too.

Author Response: Reviewer 2 seems to be sure of something they provide no evidence of. To what is this referring and where is it in the manuscript? “No explicit reference sequence for biotype 2 was available at the time of the analyses.” 

Reviewer #2: The discussion is also a very weak point on this manuscript. Some sentences and conclusions are trying to be supported by the literature that is out (or at least very far) of the subject (e.g. lines 192-195). No sound conclusions supported by the data and results arise.

Author Response: Again, this comment is opinionated conjecture and provides no real constructive criticism that can be addressed. Until there is a specific point made, there is nothing to respond to. 

Other remarks regarding the abstract and introduction.

Reviewer #2: Title: ‘biotypes’ instead of ‘biotype’ should be used since the study is based on two A. tosichella biotypes

Author Response: In the context of the title, even though the seemingly singular “biotype” is used, it is implicit that multiple biotypes are being referred to. Using the plural in this instance would be grammatically incorrect. 

Reviewer #2: Abstract: lines 35-36: something is missing in this sentence.

Author Response: The sentence reads “To date, mite-resistant wheat genotypes are the only effective method of controlling the A. tosichella - virus complex, thus the importance of elucidating A. tosichella population genetic structure, which directly affects both mite and virus management.” The sentence has been simplified to “To date, mite-resistant wheat genotypes have proven to be one of the most effective methods of controlling the A. tosichella - virus complex. Thus, it is important to elucidate A. tosichella population genetic structure, in order to better predict improved mite and virus management.” 

Reviewer #2: Lines 36-38: Previous studies have detected many more biotypes than just two.

Author Response: The text of the manuscript has been revised on lines 32-35 of the abstract and lines 74-84 of the Introduction to show that A. tosichella is a global complex of many genetic lineages.

Reviewer #2: Line 39: please use the full genus name when start the sentence from Aceria tosichella. Please check and correct the whole text considering this remark (e.g. lines 59, 70) and also please check if the Latin mite name is consequently written in italics (e.g. line 50).

Author Response: Corrected in revised manuscript. 

Reviewer #2: Lines 48-51: This conclusions suggests that biotypes 1 and 2 differ in their response to mite-resistant wheat varieties. But the study is in fact not about this, but on the biotypes occurrence and distribution. So, data presented in the manuscript did not support this conclusion.

Author Response: The authors disagree. Lines 48-51 state “The results suggest that spatio-temporal modeling can effectively improve A. tosichella management. Continual integration of precipitation and ground cover data into the existing model will further improve the accuracy of predicting the biotype composition of A. tosichella in annual wheat crops, allowing producers to make informed decisions about the selection of varieties with different A. tosichella resistance genes.” This conclusion suggests that integrating precipitation and ground cover data into the existing model will further improve accuracy of predicting A. tosichella biotype composition. 

Reviewer #2: Key words: Cmc – is not clear

Author Response: curl mite colonization (Liu et al. 2013. Crop Sci. doi:10.2135/cropsci2013.08.0564) – added to key words

Reviewer #2: Line 70-71: ‘cryptic behavior’: what do you mean?

Author Response: Cryptic is now defined as maximum concealment in the revised text.

Reviewer #2: Line 71: There are plenty of recent research that fit to be quoted here. I suggest to cite Navia et al. 2013 (https://link.springer.com/article/10.1007/s10493-012-9633-y) instead of 28 and 29. These two references are hardly to access and in fact are not adequate representative. Moreover, from several years there is an evidence that wheat curl mite is a complex of species consisting of biotypes with divergent host specificity, ranging from specialists to generalist. This should be underlined as this situation also significantly impair the detection and research on A. tosichella.

Author Response: As stated above citations 28 and 29 have been replaced with more recent publications. 

Reviewer #2: Line 76: [39] this reference is also hardly to access. Can authors here quote here to published article?

Author Response: As stated above citation 39 has been removed.

Reviewer #2: Line 79: the ‘biotype’ occurs here for the first time in the Introduction, thus the phrase ‘each biotype’ is not clear here. Authors should first introduce the readers to the issue of many biotypes identified within wheat curl mite. There is important that not only two biotypes were identified but seven in the articles that authors cite (42, 43). Further research discovered 16 and 29 biotypes respectively: https://doi.org/10.1371/journal.pone.0169874 ; https://bmcevolbiol.biomedcentral.com/articles/10.1186/s12862-018-1234-x). This issue needs to be clarified, because in the present form the background of this study lacks very important information about biology and ecology of A. tosichella and misleadingly suggests readers that only two biotypes exists. Also, authors mention (lines 80-81) that biotypes identification was made on the basis on ITS1 and COI, but ITS2 and 28S rDNA D2 regions were also used (references: 42,43).

Reviewer #2: Line 83: these references (46-48) refer to Australia and North America only, not to Europe and South America as authors state: “biotypes co-occur in mixed populations within each continent.” The reference to Europe could be: https://doi.org/10.1371/journal.pone.0169874.g002 (biotype 1 corresponds to MT-1, and biotype 2 to MT-1; you can read about this in the recent reviews (Skoracka et al. 2018a; Singh et al. 2018). 

Reviewer #2: Line 83: Change to: “At least 29 genetic lineages exist in A. tosichella complex in the world [Skoracka et al. 2018b: https://doi.org/10.1186/s12862-018-1234-x].”

Reviewer #2: Line 85: biotype 2 is mentioned in the Introduction here for the first time. Again there is necessary to develop the issue of wheat curl mite biotypes before the formulation of the objective of this study.

Author Response: Changes related to reviewer concerns about biotypes and global lineages have been incorporated in lines 32-41, and 73 -94. 

Reviewer #2: Line 91: change ‘structure’ to ‘dynamics’

Author Response: modified.

---

## [Editor Report · Decision Letter 1]

7 May 2020

Modeling Aceria tosichella biotype distribution over geographic space and time

PONE-D-19-36077R1

Dear Dr. Smith,

Thank you for submitting the revised manuscript. The changes you made in response to reviewers' comments are acceptable. Hence, I am pleased to inform you that your manuscript has been judged scientifically suitable for publication and will be formally accepted for publication once it complies with all outstanding technical requirements.

With kind regards,

Satyanarayana Tatineni, Ph.D

Academic Editor

PLOS ONE
---

## [Editor Report · Acceptance letter]

15 May 2020

PONE-D-19-36077R1 

Modeling *Aceria tosichella* biotype distribution over geographic space and time 

Dear Dr. Smith:

I am pleased to inform you that your manuscript has been deemed suitable for publication in PLOS ONE. Congratulations! Your manuscript is now with our production department. 

With kind regards,

on behalf of

Dr. Satyanarayana Tatineni 

Academic Editor

PLOS ONE